# Temporal Characterization of Behavioral and Hippocampal Dysfunction in the YAC128 Mouse Model of Huntington’s Disease

**DOI:** 10.3390/biomedicines10061433

**Published:** 2022-06-17

**Authors:** Cristine de Paula Nascimento-Castro, Elisa C. Winkelmann-Duarte, Gianni Mancini, Priscilla Gomes Welter, Evelini Plácido, Marcelo Farina, Joana Gil-Mohapel, Ana Lúcia S. Rodrigues, Andreza Fabro de Bem, Patricia S. Brocardo

**Affiliations:** 1Neuroscience Graduate Program, Center of Biological Sciences, Federal University of Santa Catarina, Florianópolis 88040-900, SC, Brazil; tine.depaula@gmail.com (C.d.P.N.-C.); gianni.mancini@gmail.com (G.M.); prii.gomesw@hotmail.com (P.G.W.); eveliniplacido@gmail.com (E.P.); marcelo.farina@ufsc.br (M.F.); ana.l.rodrigues@ufsc.br (A.L.S.R.); 2Department of Morphological Sciences, Center of Biological Sciences, Federal University of Santa Catarina, Florianópolis 88040-900, SC, Brazil; elisa.w.d@ufsc.br; 3Department of Biochemistry, Center of Biological Sciences, Federal University of Santa Catarina, Florianópolis 88040-900, SC, Brazil; andrezadebem@unb.br; 4Division of Medical Sciences and UBC Island Medical Program, University of Victoria, Victoria, BC V8P 5C2, Canada

**Keywords:** Huntington’s disease, hippocampus, YAC128 mice, electron microscopy, neurodegeneration

## Abstract

Huntington’s disease (HD) is a genetic neurodegenerative disease characterized by motor, psychiatric, and cognitive symptoms. Emerging evidence suggests that emotional and cognitive deficits seen in HD may be related to hippocampal dysfunction. We used the YAC128 HD mouse model to perform a temporal characterization of the behavioral and hippocampal dysfunctions. Early and late symptomatic YAC128 mice exhibited depressive-like behavior, as demonstrated by increased immobility times in the Tail Suspension Test. In addition, YAC128 mice exhibited cognitive deficits in the Swimming T-maze Test during the late symptomatic stage. Except for a reduction in basal mitochondrial respiration, no significant deficits in the mitochondrial respiratory rates were observed in the hippocampus of late symptomatic YAC128 mice. In agreement, YAC128 animals did not present robust alterations in mitochondrial ultrastructural morphology. However, light and electron microscopy analysis revealed the presence of dark neurons characterized by the intense staining of granule cell bodies and shrunken nuclei and cytoplasm in the hippocampal dentate gyrus (DG) of late symptomatic YAC128 mice. Furthermore, structural alterations in the rough endoplasmic reticulum and Golgi apparatus were detected in the hippocampal DG of YAC128 mice by electron microscopy. These results clearly show a degenerative process in the hippocampal DG in late symptomatic YAC128 animals.

## 1. Introduction

Huntington’s disease (HD) is the most common inherited neurodegenerative genetic disease and results from a polyglutamine expansion in the *N*-terminal region of huntingtin (htt). Even though mutant huntingtin (mHTT) is ubiquitously expressed throughout the body, it causes selective neuronal degeneration, particularly in the striatum (caudate and putamen) and specific layers of the cortex [1]. Typically, the disease is characterized by adult-onset (between 35 and 50 years of age) motor deficits, which are progressive and invariably fatal, resulting in death approximately 15–20 years later. However, the landmark of HD is the occurrence of motor symptoms (chorea, dystonia, and bradykinesia), presenting in a biphasic pattern [1]; cognitive deficits and psychiatric symptoms, such as apathy and depression, are often also present [2].

The occurrence of both cognitive deficits and psychiatric disorders in HD has traditionally been attributed to the degeneration and death of corticostriatal neurons [3]. However, increasing evidence points to the involvement of other brain regions such as the hippocampus in the etiology of these non-motor symptoms. Indeed, cell loss and reduction in the hippocampal volume have been described in patients with HD [4,5,6,7,8]. These volumetric changes in the hippocampal region have also been reproduced in animal models for HD [9,10,11,12], including the YAC128 transgenic mouse model [13]. Additionally, hippocampal neurodegeneration [14] and a reduced hippocampal neuronal area (i.e., hippocampal neuronal atrophy) [15] have also been reported in HD, and a reduction in adult hippocampal neurogenesis has been reported in the YAC128 HD mouse model [16].

Wild-type Htt is necessary for maintenance of the normal mitochondrial structure and function [17], and many studies have proposed that mitochondrial dysfunction may contribute, at least in part, to the pathogenesis of HD [17,18]. Although brain (and specifically striatal) pathology is the hallmark of HD, several studies have also indicated that peripheral tissue pathology is also present in HD and can be a contributing factor to disease progression [19,20]. In particular, skeletal muscle malfunction and HD-related cardiomyopathy have been identified in HD mouse models, and these peripheral pathological changes seemed to be accompanied by alterations in energy metabolism and mitochondrial function [21,22]. Alterations in mitochondrial biogenesis and dynamics, mitochondrial intracellular traffic, and glucose metabolism have been described in cell lines expressing mHTT, HD animal models, and HD patients [23,24,25,26]. In addition, elongated CAG repeats have been found to influence mitochondrial ATP synthesis and result in enhanced *N*-methyl-D-aspartate (NMDA) receptor activation and Ca^2+^-influx in striatal neurons [27]. The evidence of changes in striatal respiratory chain activity in HD patients also corroborates the hypothesis that mitochondrial dysfunction is implicated in HD pathophysiology [28,29,30]. However, studies that evaluated the activity of mitochondrial respiratory chain complexes in HD animal models have shown controversial results [31,32,33,34].

Although multiple pathological pathways are thought to be involved in HD, the underlying pathophysiological mechanisms responsible for the cognitive and psychiatric symptoms seen in this disorder have not yet been fully elucidated. In this study, we characterized the exploratory activity, cognition, anxiety-like, and depressive-like behaviors, as well as hippocampal mitochondrial respiratory activity, in both early (3 to 4 months) and late (11–13 months) symptomatic YAC128 mice. In addition, hippocampal morphological and ultrastructural alterations were assessed in late symptomatic YAC128 mice.

## 2. Materials and Methods

### 2.1. Animals

Adult male and female transgenic YAC128 mice (line 53) and their wild-type (WT) littermate controls were used for these experiments. The FVB/N background strain (Charles River, Senneville, QC, Canada) was used to maintain the YAC128 transgenic mouse colony, which was generated using breeding pairs generously obtain from Dr. Brian Christie (University of Victoria, Victoria, BC, Canada). At postnatal day 22, animals were weaned, ear-punched, and group-housed by sex (with a maximum of five animals per cage; cage dimensions: 19.56 × 34.70 × 14.41 cm). The colony was kept at 20–22 °C under a 12/12-h light–dark cycle (lights on at 0700 h). Mice had access to food and water *ad libitum*. All experiments were conducted between 0900 and 1600 h. All animal procedures followed the National Institutes of Health Guide for the Care and Use of Laboratory Animals and were approved by the Committee on Ethics of Animal Experimentation of the Federal University of Santa Catarina (Florianópolis, SC, Brazil; Protocol Number: 4502210318). All efforts were made to minimize animal suffering and reduce the number of mice required.

DNA extracted from ear tissue was used for genotyping of mice by conducting polymerase chain reaction (PCR) with primers for detection of YAC LYA (left YAC arm) and RYA (right YAC arm), as described in our previous study [35].

### 2.2. Experimental Protocol

For this study, YAC128 and WT mice were evaluated at two distinct stages of disease progression: early (3 to 4 months) and late (11–13 months) symptomatic. These periods were chosen based on the behavioral and neuropathological characteristics of this HD transgenic mouse model that have been previously described [36,37]. To characterize the changes observed in the early symptomatic stage, two distinct cohorts of animals were employed: cohort 1 was used to evaluate hippocampal mitochondrial function, while cohort 2 was submitted to behavioral testing. In cohort 1, mitochondrial function was assessed by high-resolution respirometry (Oroboros Oxygraph-O2K, Innsbruck, Austria) in hippocampal homogenates. In cohort 2, all animals were submitted to the following sequence of behavioral tests: the Open-Field Test (OFT), Tail Suspension Test (TST), and Swimming T-maze Test (Figure 1). Three distinct cohorts of animals were used to evaluate the alterations seen during the late symptomatic stage: cohorts 3, 4, and 5. In cohort 3, the mitochondrial function was assessed by high-resolution respirometry in hippocampal homogenates. Cohort 4 was used for hippocampal morphological and ultrastructural evaluation by electron microscopy. Finally, in cohort 5, all animals were submitted to the following sequence of behavioral tests: OFT, TST, and Swimming T-maze Test (Figure 1).

### 2.3. Behavioral Analyses

A series of behavioral analyses was conducted to assess behavioral abnormalities in early and late symptomatic YAC128 HD transgenic mice. A digital video camera (HD Pro Webcam C920 Logitech, San Francisco, CA, USA) was used to record all behavioral tests. The ANY-maze video-tracking system (Stoelting Co., Wood Dale, IL, USA) was used to analyze all behavioral data. Data analysis was performed by an investigator blinded to the genotypes of the mice.

**(a)** Open-Field Test (OFT).

The OFT was used to evaluate locomotor activity, following a previously described protocol [38]. Each individual mouse was placed in a wooden box (measurements: 40 × 60 × 50 cm). The time spent in the center of the arena, as well as the total distance traveled during a 6-min period, were recorded.

**(b)** Tail Suspension Test (TST).

Mice deprived of both acoustic and visual stimuli were suspended by their tails 50 cm above the floor for a period of 6 min. Mice were secured with adhesive tape, which was placed approximately 1 cm from the tip of their tails. The total immobility time was recorded as it has been previously described [39]. Immobility was defined as the absence of escape-related behaviors (i.e., helplessness), and mice were considered to be immobile when floating passively and motionlessly.

**(c)** Normal Phase Swimming T-maze Test.

Procedural and spatial learning was assessed using the Swimming T-maze Test [40]. In contrast with the Morris Water Maze Test, the Swimming T-maze Test does not rely on external visual cues [40], which makes it a more suitable test to test spatial learning in FVB/N mice, as these mice have been shown to develop retinal degeneration [41,42]. Clear acrylic was used to build the T-maze apparatus. This comprises three arms (30 × 7 cm): a shorter (21.5 cm long) arm and two longer (each 37 cm long) perpendicular side arms. This T-maze was filled with water (22.5 cm of water maintained at 23 ± 2 °C), and a transparent escape platform was placed in the right arm of the T-shape maze. Each mouse was placed at the base of the T-shape maze and had to learn to turn right at the top of the T so as to access the escape platform. The time and path taken to reach the escape platform were recorded. In addition, for each trial, an arbitrary score of 0 was given when mice swam to the right arm of the T-maze, whereas an arbitrary score of 1 was given when mice swam to the left arm of the T-maze. Each mouse was submitted to four trials per day (time between trials: 45 min) for a period of 3 consecutive days.

**(d)** Reversal Phase Swimming T-maze Test.

To assess strategy shifting (i.e., the ability to modify a previously learned action), the reversal phase of the Swimming T-maze Test was employed. Mice were allowed to rest for one day after completing the normal phase Swimming T-maze Test. In the following day, the escape platform was placed in the left arm of the T-maze apparatus, and mice were submitted to the reversal phase of the Swimming T-maze Test. The time and path taken to reach the escape platform, as well as the total number of arm entries, was recorded for each mouse. In addition, for each trial, an arbitrary score of 0 was given when mice swam to the left arm of the T-maze (i.e., towards the escape platform), whereas an arbitrary score of 1 was given when mice swam to the right arm of the T-maze (i.e., away from the platform). Each mouse was submitted to four trials per day (time between trials: 45 min) for a period of 3 consecutive days.

Since late symptomatic YAC128 transgenic mice present motor deficits, their swimming speed was also recorded. Three days after completion of the reversal phase Swimming T-maze Test, the stem of the T-maze apparatus was blocked, and the time taken to swim the length of the only two available arms of the T and until reaching the escape platform was recorded. Mice received five trials (time between trials: 45 min), with the last four trials used to estimate their average swimming speed.

### 2.4. High-Resolution Respirometry (Oroboros Oxygraph-O2K)

Mitochondrial function was evaluated in the hippocampus of both early (3 to 4 months) and late (11–13 months) symptomatic YAC128 HD transgenic mice and their age-matched WT controls. Mitochondrial respiratory activity was measured by determining the Oxygen Consumption Rate (OCR) using high-resolution respirometry (Oroboros Oxygraph-O2K, Innsbruck, Austria) according to the previously standardized and adapted protocols [43]. Briefly, animals were euthanized by quick decapitation, and their hippocampi were dissected and subsequently homogenized in 500 μL of respiration medium (320 mM sucrose, 1 mM EGTA, 4 mM MgCl_2_, 5 mM KH_2_PO_4_, 10 mM Tris-HCl, and H_2_O, pH 7.4) in a glass potter. After homogenization, the homogenate was pipetted into the oxygraph chamber at a concentration of 1 mg/mL of fresh tissue per chamber (total volume of each chamber: 2 mL). The substrate–uncoupler–inhibitor titration (SUIT) protocol was initiated with the determination of basal respiration (Rot), which is defined as respiration without the addition of substrates or effectors. Following respiratory stabilization, pyruvate (5 mM) and malate (0.5 mM) were added together for the evaluation of complex I (CI)-associated respiration without ATP production (Leak CI). Subsequently, ADP was added at two increasing concentrations (0.5 mM and 1 mM, respectively) to measure the OCR coupled to ATP synthesis dependent on CI (ATP/CI). Succinate (substrate for the CII, 10 mM) was then added to measure the OCR dependent on complexes I and II (ATP/CI and II). To determine the maximum rate of mitochondrial oxygen uptake by the electron transport system (ETS), titration was performed with a proton gradient dissipator, ionophore carbonyl cyanide 4-(trifluoromethoxy) phenylhydrazone (FCCP) (with subsequent injections of 0.5 μM). Finally, rotenone (0.5 μM; CI inhibitor) and antimycin (2.5 μM; CIII inhibitor) were added for total inhibition of mitochondrial OCR. The residual oxygen consumption (ROX) was measured and subtracted from the mitochondrial parameters presented in this protocol. The data acquisition and analysis of the mitochondrial respiratory rate were obtained using the DatLab software program (Oroboros Instruments, Innsbruck, Austria). The calculated parameters were Rot, Leak CI, ATP/CI, ATP/CI and II, and ETS. Representative traces of mitochondrial oxygen consumption and the main mitochondrial parameters that were calculated in this study are represented in the Results (Section 3.3). 

### 2.5. Morphological Analysis (Light and Electron Microscopy)

Since, in the YAC128 HD transgenic mouse model, both neuronal death [36,44] and striatal ultrastructural changes [45] have only been observed at an advanced stage of disease progression, in the present study, we performed a detailed morphological and structural evaluation of the hippocampal dentate gyrus (DG) only in late symptomatic (11–13 months of age) YAC128 HD mice and their WT age-matched controls (*n* = 3 to 4/genotype). Animals were anesthetized with xylazine (8 mg/kg) and ketamine (100 mg/kg) and perfused transcardially with saline containing heparin (0.1%), followed by a fixative solution (0.1 M PBS, pH 7.4, glutaraldehyde (3%), and paraformaldehyde (1.5%)). After perfusion, the brains were removed and placed in the same fixative solution for 24 h. Coronal sections (100 μm thick) were obtained with a vibratome (Vibratome, Series 1000, St. Louis, MO, USA) from 1.34 mm to 3.52 mm posterior to Bregma [46]. The sections were then washed in PBS, post-fixed in 1% osmium tetroxide (Sigma, St. Louis, MO, USA) for 2 h at room temperature, dehydrated in increasing concentrations of ethanol, and embedded in increasing concentrations of resin (Spurr resin, Electron Microscopy Sciences, Koch Instrumentos Científicos, São Paulo, SP, Brazil). For the final inclusion, sections were placed on histological slides covered with pure resin and polymerized for 48 h at 60 °C. Semithin (700 nm) sections were obtained in an ultramicrotome (Leica EMUC7, Wetzlar, Germany) and placed on slides, stained with toluidine blue (1%), and mounted with Entellan (Merck, Darmstadt, Germany). Ultrathin sections (70 nm) were cut on an ultramicrotome (Leica EMUC7), contrasted with uranyl acetate (1%) and lead citrate (2%), and viewed on an Electronic Transmitting Microscope (JEM-1011, Jeol USA Inc., Peabody, MA, USA).

### 2.6. Image Capture and Analysis

An Olympus IX83 inverted microscope (Olympus, Hamburg, Germany) coupled to an image capture system (CellSens Dimension 1.12., Olympus, Hamburg, Germany) was used for a qualitative evaluation of the semithin sections. Ultrathin sections (70 nm) were used to evaluate the DG hippocampal subregion and were examined using a JEM-1011 electronic microscope. Selected images were collected with an ES1000W Erlangshen CCD camera (GATAN Inc., Pleasanton, CA, USA) at uniform magnification, resulting in 2194 µm^2^ fields. Data collection and measurements were performed in a blinded manner. The integrity of the cell structures and the presence or absence of degenerate cells were analyzed. In addition, 15 cells without the rupture of both nuclear and cytoplasmic membranes were randomly selected from each animal (*n* = 2/genotype). Ultrastructural alterations were classified according to the following semi-quantitative classification system: − (absent), + (mild), ++ (moderate), and +++ (severe).

### 2.7. Statistical Analysis

All statistical analyses were performed using Statistica 7 software (StatSoft Inc., Tulsa, OK, USA). Two-way analyses of variance (ANOVA) were performed to determine significant differences between the genotypes (WT and YAC128) and stages of disease progression (early and late symptomatic), followed by Duncan’s multiple range post hoc test when appropriate. Repeated measures ANOVA and Student’s *t*-tests were used to compare WT and YAC128 mice during motor training and spatial learning. Comparisons of categorical data were performed with a chi-square test with Yates’s correction when necessary. Data were expressed as the means ± SEM and considered significant when *p* < 0.05.

## 3. Results

### 3.1. Effect of Genotype and Stage of Disease Progression on Locomotor Activity and on the Occurrence of Depressive-Like Behaviors

Twelve-months old YAC128 mice showed a decrease in locomotor activity in the OFT as compared to YAC128 mice at 4 months of age (*p* < 0.05) (Figure 2a). In addition, 4-months old YAC128 mice spent less time in the center of the OFT when compared to their age-matched WT counterparts (*p* < 0.05) and as compared with 12-month-old YAC128 mice (*p* < 0.01) (Figure 2b).

In the TST, YAC128 mice showed a longer immobility time at 4 (*p* < 0.01) and 12 (*p* < 0.05) months of age when compared to their age-matched WT controls. In addition, at 12 months of age, transgenic YAC128 mice showed a significant reduction in immobility time when compared to younger YAC128 mice in this behavioral test (*p* < 0.01) (Figure 2c).

### 3.2. Effect of Genotype and Stage of Disease Progression on Procedural and Spatial Learning

The swimming T-maze test was used to evaluate procedural and spatial learning [40] in early and late symptomatic YAC128 transgenic mice. After four training trials, early and late symptomatic YAC128 mice and their age-matched WT controls showed a decrease in the time to reach the platform (Figure 3a). The path traveled by each animal was also evaluated. Swimming right (platform) was arbitrarily given a score of 0, whereas swimming left was given a score of 1. On the first day of the normal phase of the T-maze Test (first 4 trials), 12-month-old YAC128 mice made more errors than the 4-month-old YAC128 group (X^2^ = 8.40). On the second day, all groups had a similar performance. On the third day, 12-month-old YAC128 mice made more errors when compared with their age-matched WT controls (X^2^ = 4.12) and with 4-month-old YAC128 mice (X^2^ = 5.06) (Figure 3b).

The platform was switched from the right arm to the left arm of the T-maze to perform the reversal phase of the Swimming T-maze Test and assess the ability of YAC128 mice to change strategy. In the first trial, after switching the platform to the left arm, 12-month-old YAC128 mice took longer to reach the platform (Figure 3c). On the second day of the reversal phase, 4-month-old WT (X^2^ = 16.29) and YAC128 (X^2^ = 17.06) mice showed better performances when compared with their respective 12-month-old genotype groups (Figure 3d). Similar results were found on the third day of the reversal phase, when 12-month-old WT mice made significantly more mistakes to reach the platform when compared to their 4-month-old WT counterparts (X^2^ = 4.49). Moreover, 12-month-old YAC128 mice made significantly more errors than their 4-month-old YAC128 counterparts (X^2^ = 14.00) (Figure 3d).

These findings suggest that both age and genotype may contribute to the cognitive deficits seen in 12-month-old WT and YAC128 mice. Of note, to discard the influence of motor deficits on the T-maze results, swimming speeds were calculated, and no significant differences between genotypes were observed (WT = 24.44 cm/s; YAC128 = 25.70 cm/s; *p* = 0.37).

### 3.3. Effect of Genotype and Stage of Disease Progression on the Evaluation of Mitochondrial Oxygen Consumption in the Hippocampus

Mitochondrial respiratory activity was evaluated on hippocampal homogenates from both early and late symptomatic YAC128 HD transgenic mice and their age-matched WT controls using the OROBOROS 2K Oxymeter. A representative trace of mitochondrial O_2_ consumption and the main mitochondrial parameters evaluated are illustrated in Figure 4a. Early and late symptomatic YAC128 mice had similar hippocampal respiratory rates under all test conditions as compared with their age-matched WT counterparts. Although a two-way ANOVA demonstrated a significant main effect of the genotype (F(1,18) = 4.82, *p* = 0.04) with YAC128 mice showing a significant reduction in mitochondrial respiratory activity compared with WT mice, no significant main effect of the symptomatic stage (F(1,18) = 0.36, *p* = 0.55), and no significant interaction between genotype and symptomatic stage were observed (F(1,18) = 2.41, *p* = 0.13) (Figure 4b).

### 3.4. Effect of Genotype on the Morphology of the Hippocampal DG in Late Symptomatic YAC128 Mice

In the hippocampal DG of WT mice (Figure 5a), only typical neurons, with neuronal cell bodies exhibiting normal shape and staining intensity and rounded pale nuclei, were observed (Figure 5b). On the other hand, darkly stained granule cell bodies with irregular nuclei, surrounded by a thin rim of deeply stained cytoplasm (shrunken nuclei and cytoplasm) and abnormal stained intensity (red arrows), were detected in the hippocampal DG of YAC128 mice (Figure 5c).

### 3.5. Effect of Genotype on the Ultrastructure of the Hippocampal DG in Late Symptomatic YAC128 Mice

The electron microscopy analysis of WT DG showed a densely packed granule cell layer (Figure 6A) with intact granule cell bodies and a large, pale, oval, rounded nucleus (abundance of euchromatin or uniformly dispersed chromatin). Figure 6B, a higher magnification view of the boxed area in Figure 6A, shows a granule cell with a normal appearance. A conserved nuclear and plasma membrane (red arrow) were seen, and heterochromatin margination in the nuclear membrane and/or clusters dispersed in the nucleus (arrowhead) were also noted. In Figure 6C, some cellular structures can be observed in more detail, such as the nucleus (N), the Golgi apparatus (G), the rough endoplasmic reticulum (ER), and the mitochondria (M).

On the other hand, a qualitative examination revealed several alterations in the morphology of DG neurons in YAC128 mice (Figure 6D). Indeed, in the DG of transgenic mice, it was possible to visualize dark neurons with an irregular outline and shrunken nuclei (red arrows). With regards to ultrastructural alterations, granule cells with an intact nucleus and cytoplasm with normal staining (normal electron density) but with altered (i.e., dilated) organelles were visible (Figure 6E). More specifically, granule cells exhibiting dilated ER segments and an enlarged Golgi apparatus with swollen terminal cisternae were detected. However, most mitochondria showed normal profiles, while fewer had a mild crest loss. Importantly, these ER and Golgi apparatus changes were seen in all the YAC128 DG cells evaluated (Figure 6F).

Of note, only intact DG granule cells with well-delimited nuclear and plasma membranes were chosen for the ultrastructural analyses. The ultrastructural alterations that were seen in the WT and YAC128 DG granule cells are summarized in Table 1.

## 4. Discussion

In this study, we characterized the cognitive and affective deficits in both early (3 to 4 months) [37,40] and late (11–13 months) [36,47,48] symptomatic YAC128 HD mice and assessed whether these behavioral abnormalities were accompanied by deficits in mitochondrial function or morphological and ultrastructural alterations in the hippocampus of these HD mice.

Depression is the most common psychiatric disturbance seen in HD individuals and is often present years before the onset of motor and cognitive changes [49,50]. In this study, YAC128 mice exhibited a depressive-like behavior at both the symptomatic stages analyzed, as shown by a significant increase in the immobility time in the TST at 4 and 12 months of age. This agrees with previous evidence showing that depressive-like behavior can be seen in the YAC128 HD mouse model as early as 2 months of age and remain present as the disease progresses [51,52]. Furthermore, psychiatric symptoms during the early symptomatic stage have also been observed in other HD transgenic mouse models [53,54,55,56,57,58]. Together, these studies mimic the phenotypes observed in individuals with HD, who show psychiatric changes such as depression early in the progression of the disease [59,60,61].

Clinically, anxiety is also a common feature seen in HD but is often poorly investigated. The prevalence of anxiety in individuals with HD can range from 13 to 71%, depending on the study. This psychiatric disorder can be present at different stages of the disease and is often associated with depression and irritability [62]. In the present study, anxiety-related behavior was evaluated by assessing the time spent in the central region of the open field. This parameter is inversely related to the degree of anxiety [63]. At 4 months of age, YAC128 mice showed a reduction in the time spent in the central area of the open field. However, this behavior was not present when animals reached 12 months of age, indicating that this disturbance may be characteristic of the early symptomatic phase of disease progression. In line with this result, Chiu et al. (2011) showed a reduction in the time spent in the center of the open field in YAC128 mice at 6 months of age [64]. Of note, in our previous study, 3-month-old YAC128 mice did not display anxiety-related behavior [35], a finding that was in agreement with the study from Pouladi et al. (2008), who also showed no changes in anxiety-related behaviors (as assessed by the hyperthermia-induced anxiety test) in YAC128 mice at both 3 and 4 months of age [37]. These discrepancies among the studies might be related, at least in part, to the fact that different behavioral tests have been used to assess the occurrence of anxiety-related behaviors in YAC128 mice [35,64,65,66]. Nevertheless, although some variance may occur regarding the exact age at which anxiety-like behaviors occur in the YAC128 HD mouse model (and in the human condition), collectively, these studies indicate that anxiety-related behaviors may indeed be part of the phenotype observed during the early symptomatic stage of this disease.

Several studies have shown that cognitive deficits can precede the onset of motor symptoms (i.e., clinical diagnosis of the disease) in HD patients. In the early stages, individuals may present with deficits in strategy change, psychomotor speed, spatial recognition memory, planning, and verbal fluency [67,68,69,70]. In the YAC128 HD mouse model, a deficit in motor learning was detected as early as 2 months of age, while impairments in learning and memory, spatial memory, and cognitive flexibility were observed at 8 and 12 months of age [40].

Behavioral tests that assess hippocampal-dependent learning and memory (e.g., spatial memory), such as the Swimming T-maze Test, have been previously employed in transgenic models for HD [40,71]. In the present study, animals were submitted to consecutive training sessions to promote learning of the task (defined by a reduction in the time to reach the target platform and a decrease in the number of entries into the wrong arms of the T-maze). On the first day of the test, 12-month-old YAC128 animals spent significantly more time reaching the target platform when compared to the 4-month-old YAC128 mice. However, in subsequent test days, a reduction in the latency time to reach the platform was observed for both genotypes and age groups. Nevertheless, although no significant changes in the time taken to reach the platform were detected between YAC128 and WT mice, YAC128 mice exhibited a significant increase in the number of entries into the wrong arms of the T-maze. By the end of the test, almost all WT mice had learned the correct path to the platform, whereas YAC128 animals tended to swim to the opposite side or return to the base of the T. This difference in spatial response strategy can be interpreted as an impairment in spatial navigation, which is in line with cognitive dysfunction being part of the phenotype of this HD transgenic mouse model. During the reversal phase of the Swimming T-maze Test, when the ability to change strategy was assessed, an effect of age was found, with 4-month-old WT and YAC 128 mice performing better than their 12-month-old counterparts.

To date, most studies have evaluated mHTT-induced changes in mitochondrial function in the striatum, the brain structure most affected in HD. To the best of our knowledge, there are currently no studies investigating mitochondrial function in the hippocampus of HD mouse models. In the present study, mitochondrial function was evaluated in hippocampal homogenates from YAC128 and WT mice using high-resolution respirometry. Our results show that mitochondrial oxygen consumption in the hippocampus was similar between WT and YAC128 mice both at the early and late stages of disease progression, demonstrating that hippocampal mitochondrial respiration was not altered in YAC128 HD transgenic mice at the ages tested and with the protocol used. However, it is possible that more subtle alterations at the level of the expression of individual mitochondrial respiratory chain enzymes and/or with regards to AMP/ADP/ATP levels may still be present in the hippocampus of YAC128 mice. As such, future studies are warranted to further elucidate hippocampal mitochondrial changes in this HD transgenic mouse model. Nevertheless, and in alignment with our results, Hamilton et al. (2015 and 2019) have also failed to detect significant differences in the brain mitochondrial respiratory activity between YAC128 and WT at 2, 4, and 10 months of age [33,34]. Additionally, Yano et al. (2014) reported no alterations in the brain mitochondrial respiratory function in R6/2 mice, regardless of the length of their CAG repeat expansion (150 CAGs or 195 CAGs) or stage of disease progression (pre-symptomatic phase at 5 to 6 weeks of age or advanced symptomatic stage at 10 to 11 weeks of age) [32]. Moreover, although the respiratory activity was shown to be reduced in isolated mitochondria from the striatum of R6/1 mice, the same was not observed in isolated mitochondria from the cortex of these mice [72]. Furthermore, no alterations in the activity levels of complexes II, III, and IV were found in mitochondria isolated from whole brains of transgenic N171-82Q mice [73]. Similarly, no changes in the activity levels of complexes I, II, III, and IV have been detected in the striatum and cerebral cortex of HD48 and HD49 transgenic mice [31]. Finally, no significant change in the mitochondrial complex II subunit activity was detected in an in vitro HD model (STHdhQ111/Q111 cells) [74].

While results with the postmortem tissue of individuals with HD appear to be consistent with mitochondrial dysfunction [28,29,30], the subtle changes in respiratory rates in hippocampal homogenates observed in the present study suggest that mHTT does not greatly impact the overall mitochondrial oxidative metabolism in the hippocampus of the YAC128 HD transgenic mouse model. Therefore, it is unlikely that mitochondrial respiratory dysfunction plays a prominent role in the etiology of the behavioral changes described in the present study and the hippocampal neuropathological changes seen in the early and advanced symptomatic stages of disease progression in YAC128 HD mice. However, we cannot rule out the possibility that mitochondrial dysfunction may play a more prominent role in the more advanced/late stages of disease progression. Still, in this case, such dysfunction probably represents a consequence of the evolving pathology rather than a triggering event for the neurodegeneration present in this brain region.

The present study also assessed whether morphological and ultrastructural changes could be detected in the hippocampus of late symptomatic YAC128 HD transgenic mice. Using hippocampal semithin sections, we found evidence of neuronal death in the hippocampal DG of 12-month-old YAC128 mice. Neurodegeneration was characterized by hyperchromic neurons with irregular cell morphology and condensed nuclei, revealing cell retraction. Previous studies have reported similar morphological alterations in the striatum of different HD transgenic mouse models at advanced stages of disease progression, including R6/1 [75], R6/2 [76,77], and BACHD [78] transgenic mice. Of note, although one previous study reported morphological changes in hippocampal CA3 pyramidal cells in TgN6/1 mice [14], to the best of our knowledge, the present study has been the first to provide a detailed characterization of the morphological (and ultrastructural) changes in the hippocampal DG of an HD transgenic mouse model.

Nevertheless, previous studies have demonstrated ultrastructural alterations in cortical and striatal neurons of various HD transgenic mouse models and human HD patients. Indeed, postmortem studies have revealed the presence of hyperchromic degenerate neurons with condensed nuclei and retraction of the plasma membrane in the anterior cingulate cortex [77] and nuclear indentations in neurons of the nucleus accumbens, caudate nucleus, and frontal cortex [79], as well as Golgi apparatus and membrane alterations, crest-impaired mitochondria, disorganized ER, and condensed chromatin in the frontal cortex [80] of HD patients. Yamanishi et al. (2017) described cortical neurons with ER dilation without the presence of autophagosomes or apoptotic bodies in Htt-KI mice (as well as HD patients) [81]. Bayram-Westom et al. (2012) reported intense vacuolization and the degeneration of cytoplasmic organelles in striatal cells from YAC128 mice at 14 months of age [45], while Mastroberardino et al. (2002) observed dilated ER in the striatum and cerebral cortex of R6/1 mice [75]. Stack et al. (2005) found abnormalities in the mitochondrial and ER morphology in the striatum of R6/2 mice [76]. However, no studies have evaluated the ultrastructure of hippocampal DG granule cells in transgenic HD mouse models. The ultrastructural analysis performed in the present study demonstrated the presence of DG granular cells with a loss of plasma membrane integrity, dark cytoplasm and reduced cytoplasmic density, scarcity of cytoplasmic organelles, nuclear retraction, and little chromatin condensation in the hippocampus of late symptomatic YAC128 mice. Although no vacuoles or apoptotic bodies were found in YAC128 DG granular cells, presented with prominent dilation of their rough ER and Golgi apparatus and only slight changes in their mitochondrial morphology (further corroborating the findings of normal mitochondrial respiratory activity reported here and discussed above). Although the classification criteria between apoptosis and necrosis have not been well-defined in HD [81], the alterations observed in the YAC128 hippocampal DG reported here appear to be similar to the ultrastructural changes commonly associated with necrotic cell death.

## 5. Conclusions

The temporal characterization of the behavioral deficits and hippocampal biochemical and morphological alterations seen in YAC128 transgenic mice reported here further contribute to the characterization of disease progression in this HD transgenic mouse model. In particular, the present study has further characterized the mood and cognitive changes seen in early and late symptomatic YAC128 mice. Of note, the behavioral alterations reported here were accompanied by subtle changes in hippocampal mitochondrial function, suggesting that this mechanism does not play a major role in the mood and cognitive changes observed in both early and late symptomatic YAC128 mice. However, a prominent neurodegenerative process was detected in hippocampal DG granule cells from late symptomatic YAC128 mice. Such hippocampal neurodegeneration may contribute, at least in part, to the deficits in hippocampal-dependent behaviors seen in late symptomatic YAC128 mice and, potentially, individuals afflicted with this devastating disorder.

## Figures and Tables

**Figure 1 biomedicines-10-01433-f001:**
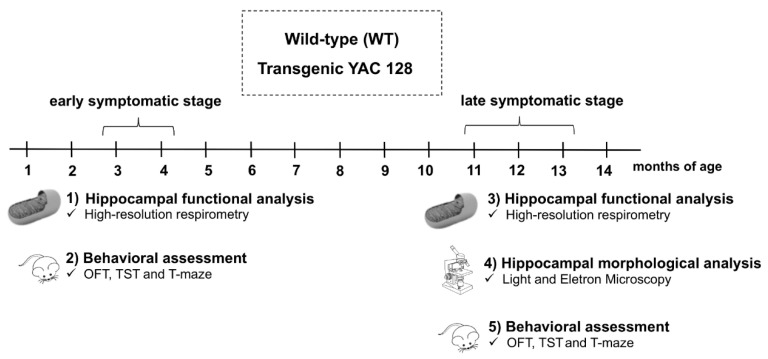
Experimental protocol. Adult male and female WT and YAC128 mice were evaluated at two distinct stages: early (3 to 4 months of age) and late (11–13 months of age) symptomatic. Animals were divided into five cohorts. For behavioral analyses, animals were submitted to the Open-Field Test (OFT), the Tail Suspension Test (TST), and T-maze Swimming Test. Hippocampal mitochondrial function using high-resolution respirometry was performed at both the early and late symptomatic stages, while hippocampal morphological and ultrastructural analyses were only performed at the late symptomatic stage.

**Figure 2 biomedicines-10-01433-f002:**
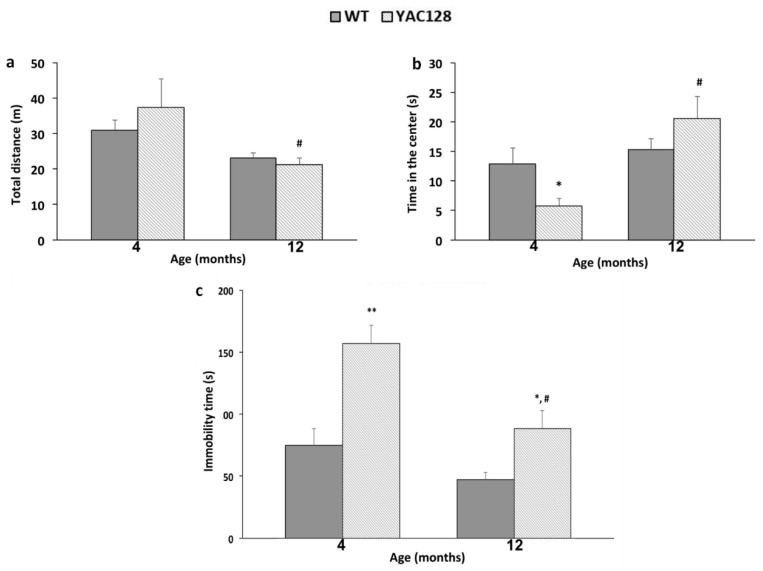
Effects of the genotype and stage of disease progression on the locomotor activity and motor coordination in WT and YAC128 mice, as assessed with the Open-Field Test (OFT). Distance traveled (**a**) and time spent in the center (**b**) were the parameters evaluated in the OFT. (**c**) Evaluation of depressive-like behavior in WT and YAC128 mice assessed by the Tail Suspension Test (TST) at 4 and 12 months of age. The total immobility time is represented as the mean ± SEM. The results were analyzed with two-way ANOVA, followed by Duncan’s multiple range post hoc test. *n* = 7–10 mice/group. * *p* < 0.05 and ** *p* < 0.01 when compared to the WT group; ^#^
*p* < 0.05 when compared to the 4-month-old YAC128 animals.

**Figure 3 biomedicines-10-01433-f003:**
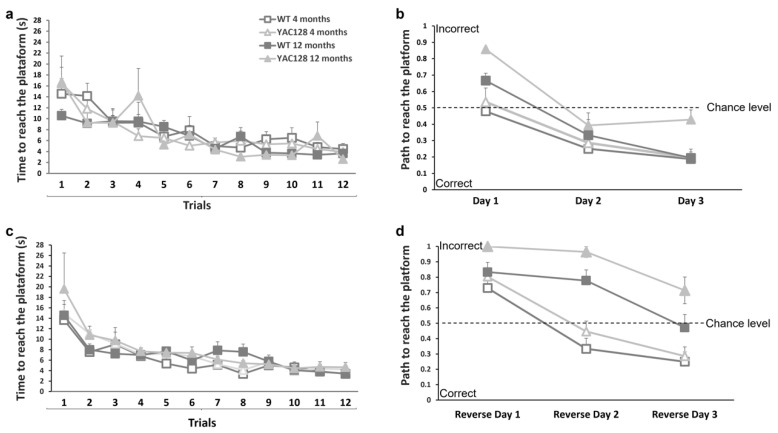
Assessment of spatial learning in early and late symptomatic YAC128 mice and age-matched WT controls. Animals were trained to swim to a platform located in the right arm of a T-maze pool. After 3 days of four trials each day, the platform was switched to the left arm of the maze. The time to reach the platform (**a**,**c**) and the platform’s path (**b**,**d**) were recorded. Values represent the means ± SEM. Repeated measures ANOVA and chi-square test with Yates’s correction were used to detect statistical differences, *n* = 7–9.

**Figure 4 biomedicines-10-01433-f004:**
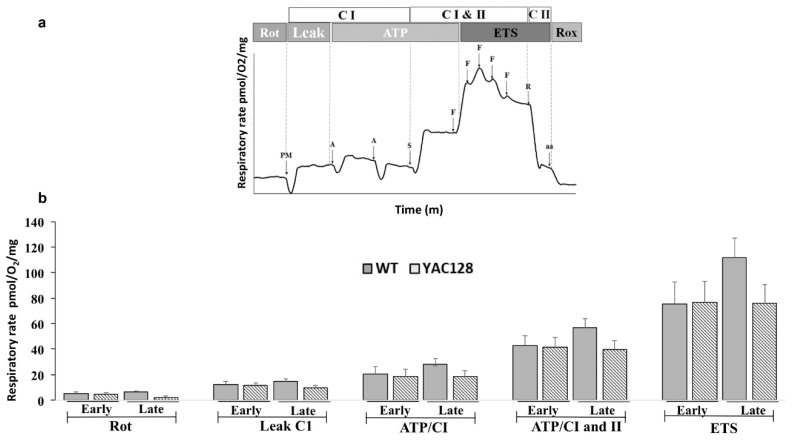
Mitochondrial respiratory activity in hippocampal homogenates from early (3 to 4 months) and late (11–13 months) symptomatic YAC128 mice and their age-matched WT counterparts. Representative traces of the main mitochondrial parameters calculated (**a**) and the respiratory states from hippocampal homogenates in WT and YAC128 mice (**b**). Rot—the rate of basal O_2_ consumption, Leak—O_2_ consumption rate associated with Complex I independent of ADP, ATP/CI and ATP/CI and II—the rate of O_2_ consumption associated with ATP production-related only to complex I (ATP/CI) and to complex I and II (ATP/CI and II), respectively, and ETS—maximal mitochondrial respiration. Statistical analysis of the respiratory rates was performed using two-way ANOVAs, and the values were expressed as the means ± SEM. Abbreviations: CI—complex I, CII—complex II, PM—pyruvate and malate, A—adenosine diphosphate (ADP), S -succinate, F—carbonyl cyanide-4-(trifluoromethoxy)phenylhydrazone (FCCP), R—rotenone, and aa—antimycin. *n* = 5 to 6.

**Figure 5 biomedicines-10-01433-f005:**
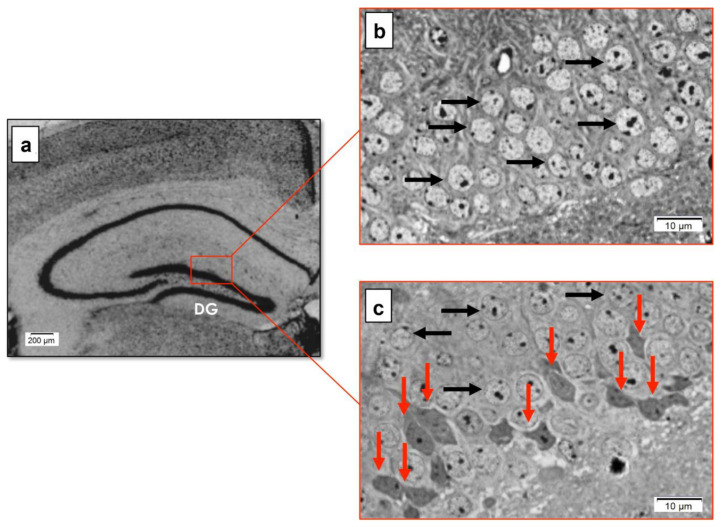
Qualitative morphological analyses of the hippocampal DG region in late symptomatic YAC128 mice and age-matched WT controls. Photomicrography of a coronal hippocampal section from age-matched WT mice (**a**). Representative semithin DG sections from WT (**b**) and YAC128 (**c**) mice. Typical granule cell bodies with a normal shape and regular staining intensity are present in the hippocampal DG of WT and YAC128 mice (black arrows) (**b**,**c**). Dark granule cells appearing shrunken, with an irregular outline and increased staining intensity (red arrows), can also be noted in the hippocampal DG of YAC128 mice (**c**).

**Figure 6 biomedicines-10-01433-f006:**
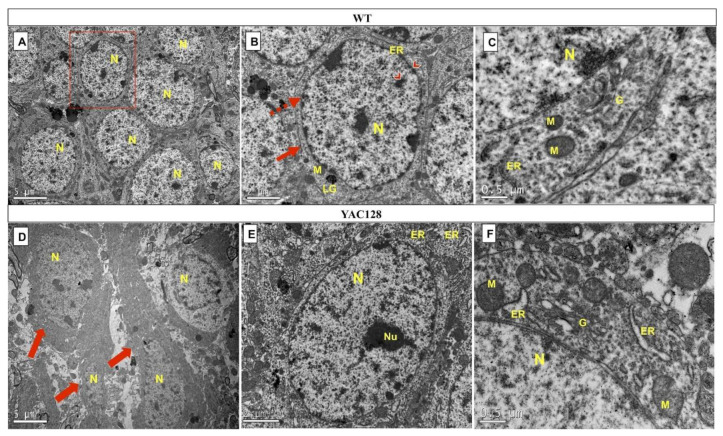
Ultrastructural analysis of DG granular neurons from late symptomatic YAC128 mice and age-matched WT controls. Granule cells bodies from WT mice can be seen in the upper panel (**A**–**C**) compared with YAC128 mice in the bottom panel (**D**–**F**). Higher magnification view of the boxed area in (**A**), representing a typical WT DG granule neuron with preserved nuclear (dashed arrow) and plasma (straight arrow) membranes. Note the heterochromatin margination in the nuclear membrane and some clusters dispersed in the nucleus (red arrow) (**B**). WT DG granule neuron with normal shape and normal electron density of the cytoplasmic and nuclear membrane and intact cytoplasmic organelles (ER, M, and G) (**C**). In contrast, YAC128 DG granule cells are more electron-dense (both with regards to their cytoplasm and nuclei), as evidenced by the presence of dark neurons, which exhibit an irregular outline and shrunken nucleus and cytoplasm (open arrows) (**D**). YAC128 intact DG granule cells with a conserved nucleus and cytoplasmic shape, but dilated organelles can also be observed (**E**). Dilated ER and enlarged Golgi cisternae were seen, although mitochondria appear normal or show a mild crest loss (**F**). Abbreviations: G, Golgi apparatus; LG, lipofuscin granules; M, mitochondria; N, nucleus; ER, rough endoplasmic reticulum (scale bar = 5 µm (**A**,**D**), 2 µm (**B**,**E**), and 0.5 µm (**C**,**F**)).

**Table 1 biomedicines-10-01433-t001:** Ultrastructural alterations were seen in DG granular neurons of WT and YAC128 mice in the late stage of HD.

Cell Structures	WT	YAC128
Nucleus	**-**	**+**
Nuclear membrane	**-**	**+**
Plasma membrane	**-**	**+**
Cytoplasmic organelles:		
Mitochondria	**-**	**+**
Golgi apparatus	**+**	**++**
Rough endoplasmic reticulum	**-**	**+++**
Lipofuscin granules	**-**	**-**

Ultrastructural changes in DG granular neurons were classified as - (absent), + (mild), ++ (moderate), and +++ (severe).

## Data Availability

The data presented in this study are available within the article text and figures.

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
