# Peer review of "Temporal Characterization of Behavioral and Hippocampal Dysfunction in the YAC128 Mouse Model of Huntington’s Disease"

_biomedicines, 2022, doi:10.3390/biomedicines10061433_

Round 1

Reviewer 1 Report

  1. In the manuscript, the authors investigated the behavioral changes, hippocampal dysfunction as well as changes in mitochondrial functionality in one of HD mice model. Although the article is generally well written and covers main topics, there are issues, the authors need to consider.

    1. This topic and this kind of studies were published many times during the years of HD research thus in my opinion authors should strongly consider to add some more experiments focused on confirmation of no changes in mitochondrial functionality of hippocampus (described below) or focus on other interesting finding and add some other aspects of hippocampal changes.
    2. In the introduction section, the authors should :
    • mentioned the nowadays HD is considered also as a multi-system disorder mainly due to skeletal muscle and heart function derangements, in those systems energy metabolism and mitochondrial changes were also noted,
    • highlighted the role of mitochondria in calcium Ca2+ buffering which is crucial for HD mitochondrial dysfunctionality and is linked with NMDA receptors in the striatum
    • add the information about HTT and mTT role in mitochondrial functionality.
    1. In the results section, the authors are very much welcome to consider adding and measure the transcript and levels of protein involved in ETC. Because overall ECT functionality depends on successful gene transcription, then translation and protein level as well as enzyme activity which could also be deregulated by mHTT aggregates.  Moreover, to provide a complete evaluation of ECT authors should consider measure also the coenzyme Q, SDH or citric synthase levels in investigated brain region. Also, the information about ATP levels in hippocampus will be valuable to evaluate.
    2. Please take a look into some works:
      1. “Purine Nucleotides Metabolism and Signaling in Huntington’s Disease: Search for a Target for Novel Therapies”. International Journal of Molecular Sciences. 2021; 22(12):6545. https://doi.org/10.3390/ijms22126545
      2. “Skeletal muscle pathology in Huntington’s disease.” Front. Physiol. 2014, 5, 380. 
      3. “Huntingtin protein maintains balanced energetics in mouse cardiomyocytes”. Nucleosides Nucleotides Nucleic Acids 2020,1–8.
      4. “Huntingtin protein is essential for mitochondrial metabolism, bioenergetics and structure in murine embryonic stem cells.” Dev. Biol. 2014, 391, 230–240.
      5. “Huntington's disease is a multi-system disorder”. Rare Dis. 2015 Jul 24;3(1):e1058464. doi: 10.1080/21675511.2015.1058464. PMID: 26459693; PMCID: PMC4588536.

Author Response

Reviewer #1

In the manuscript, the authors investigated the behavioral changes, hippocampal dysfunction as well as changes in mitochondrial functionality in one of HD mice model. Although the article is generally well written and covers main topics, there are issues, the authors need to consider. This topic and this kind of studies were published many times during the years of HD research thus in my opinion authors should strongly consider to add some more experiments focused on confirmation of no changes in mitochondrial functionality of hippocampus (described below) or focus on other interesting finding and add some other aspects of hippocampal changes.

  1. In the introduction section, the authors should:
  • mentioned the nowadays HD is considered also as a multi-system disorder mainly due to skeletal muscle and heart function derangements, in those systems energy metabolism and mitochondrial changes were also noted.

Response: Please note that we have now added the following paragraph to the Introduction Section, so as to address this suggestion:

Although brain (and specifically striatal) pathology is the hallmark of HD, several studies have also indicated that peripheral tissue pathology is also present in HD and can be a contributing factor to disease progression [19,20]. In particular, skeletal muscle malfunction and HD-related cardiomyopathy have been identified in HD mouse models, and these peripheral pathological changes seemed to be accompanied by alterations in energy metabolism and mitochondrial function (21,22).

  • highlighted the role of mitochondria in calcium Ca2+ buffering which is crucial for HD mitochondrial dysfunctionality and is linked with NMDA receptors in the striatum

Response: Please note that we have now added the following paragraph to the Introduction Section, so as to address this suggestion:

In addition, elongated CAG repeats have been found to influence mitochondrial ATP synthesis and result in enhanced N-methyl-D-aspartate (NMDA) receptor activation and Ca2+-influx in striatal neurons (27). Evidence of changes in striatal respiratory chain activity in HD patients also corroborates the hypothesis that mitochondrial dysfunction is implicated in HD pathophysiology [28–30].

  • add the information about HTT and mTT role in mitochondrial functionality.

Response: We have now added the following sentence to the revised manuscript. Please note that this sentence includes citations to studies that have evaluated the role of Htt in maintaining mitochondrial structure and function:

“Wild-type Htt is necessary for maintenance of normal mitochondrial structure and function [17] and many studies have proposed that mitochondrial dysfunction may contribute, at least in part, to the pathogenesis of HD [17, 18].”

  1. In the results section, the authors are very much welcome to consider adding and measure the transcript and levels of protein involved in ETC. Because overall ECT functionality depends on successful gene transcription, then translation and protein level as well as enzyme activity which could also be deregulated by mHTT aggregates.  Moreover, to provide a complete evaluation of ECT authors should consider measure also the coenzyme Q, SDH or citric synthase levels in investigated brain region. Also, the information about ATP levels in hippocampus will be valuable to evaluate.

Response: We agree with the Reviewer that it would have been interesting to further investigate the involvement of mitochondrial dysfunction (e.g., expression of ETC proteins) in YAC128 HD mice. However, unfortunately, we no longer possess tissue samples to perform these additional experiments and given the short time provided by the Journal to resubmit our manuscript, we are unable to generate additional samples at this time (please note that it would take several months to be able to collect new samples from late-symptomatic YAC 128 mice and their wild-type age-matched controls). However, the preliminary assessment of the mitochondrial ETC included in the present study will constitute the focus of our future studies. Moreover, we have now discussed this pitfall in the Discussion Section, by mentioning that while we did not find alterations in mitochondrial function in this study, other alterations at the level of individual enzyme expression and/or AMP/ADP/ATP levels may still be present and contribute to HD pathology. This paragraph reads as follows:

However, it is possible that more subtle alterations at the level of the expression of individual mitochondrial respiratory chain enzymes and/or with regards to AMP/ADP/ATP levels may still be present in the hippocampus of YAC128 mice. As such, future studies are warranted to further elucidate hippocampal mitochondrial changes in this HD transgenic mouse model.

  1. Please take a look into some works:
    • (i) “Purine Nucleotides Metabolism and Signaling in Huntington’s Disease: Search for a Target for Novel Therapies”. International Journal of Molecular Sciences. 2021; 22(12):6545. https://doi.org/10.3390/ijms22126545
    • (ii) “Skeletal muscle pathology in Huntington’s disease.” Front. Physiol. 2014, 5, 380. 
    • (iii) “Huntingtin protein maintains balanced energetics in mouse cardiomyocytes”. Nucleosides Nucleotides Nucleic Acids 2020,1–8.
    • (iv) “Huntingtin protein is essential for mitochondrial metabolism, bioenergetics and structure in murine embryonic stem cells.” Dev. Biol.2014, 391, 230–240.
    • (v) “Huntington's disease is a multi-system disorder”. Rare Dis. 2015 Jul 24;3(1):e1058464. doi: 10.1080/21675511.2015.1058464. PMID: 26459693; PMCID: PMC4588536.

Response: We have now cited most of these suggested references in the revised version of the Introduction Section as well as some additional references that support our answers to the points raised by the reviewer (e.g., Reddy et al. 2009 – Brain Res. Rev. and Björkqvist et al. 2005 – Hum. Mol. Genet.).

Reviewer 2 Report

The authors present an extensive analysis of a HD model in the hippocampus to investigate this brain structure's role in HD pathophysiology. A few comments to consider:

-was randomization used for the mice that were sacrificed at different time points for the analyses?

-page 7, last line; is the p-value of 2.41 correct?

-given the seeming lack of effect in your high-res respirometry analysis, should other mitochondrial pathway analyses be considered? What is the sensitivity of this approach to detect effects in this pathway?

Author Response

Reviewer #2

The authors present an extensive analysis of a HD model in the hippocampus to investigate this brain structure's role in HD pathophysiology. A few comments to consider:

  1. Was randomization used for the mice that were sacrificed at different time points for the analyses?

Response: This is correct. Mice were randomly assigned to each age-group using an Excel spreadsheet.

  1. Page 7, last line; is the p-value of 2.41 correct?

Response: We thank the Reviewer for pointing out this typographical error. This has now been corrected in the revised manuscript. The correct sentence reads as follows:

“… no significant interaction between genotype and symptomatic stage were observed [F (1,18) = 2.41, p = 0.13].”

  1. Given the seeming lack of effect in your high-res respirometry analysis, should other mitochondrial pathway analyses be considered? What is the sensitivity of this approach to detect effects in this pathway?

Response: As per our response to Reviewer #1, Question #2, we agree that it would have been interesting to further investigate the involvement of mitochondrial dysfunction (e.g., expression of ETC proteins) in YAC128 HD mice. However, unfortunately, we no longer possess tissue samples to perform these additional experiments and given the short time provided by the Journal to resubmit our manuscript, we are unable to generate additional samples at this time (please note that it would take several months to be able to collect new samples from late-symptomatic YAC 128 mice and their wild-type age-matched controls). However, the preliminary assessment of the mitochondrial ETC included in the present study will constitute the focus of our future studies. Moreover, we have now discussed this pitfall in the Discussion Section, by mentioning that while we did not find alterations in mitochondrial function in this study, other alterations at the level of individual enzyme expression and/or AMP/ADP/ATP levels may still be present and contribute to HD pathology. This paragraph reads as follows:

However, it is possible that more subtle alterations at the level of the expression of individual mitochondrial respiratory chain enzymes and/or with regards to AMP/ADP/ATP levels may still be present in the hippocampus of YAC128 mice. As such, future studies are warranted to further elucidate hippocampal mitochondrial changes in this HD transgenic mouse model.

With regards to the sensitivity of the method employed, the Oroboros Oxygraph-O2K used in our study is based on a high-resolution design that maximizes respirometric sensitivity and precision (minimal O2 leak and highly sensitive electrodes), reducing the biological sample size required [Mitochondrial Physiology Network 18.10: 1-8 (2014)].

Round 2

Reviewer 1 Report

Thanks to the authors for manuscript corrections and for appreciating reviewer tips.